# Epidemiology of *Streptococcus pneumoniae* Serotypes in Jordan Amongst Children Younger than the Age of 5: A National Cross-Sectional Study

**DOI:** 10.3390/vaccines11091396

**Published:** 2023-08-22

**Authors:** Munir Abu-Helalah, Asma’a Al-Mnayyis, Hamed Alzoubi, Ruba Al-Abdallah, Hussein Jdaitawi, Omar Nafi, Kamel Abu-Sal, Alaa Altawalbeh, Alia Khlaifat, Enas Al-Zayadneh, Ihsan Almaaitah, Ibrahim Borghol, Fadi Batarseh, Omar Okkeh, Abdallah Dalal, Ahmad Alhendi, Mohammad Almaaitah, Adnan Al-Lahham, Mahmoud Gazo, Faisal Abu Ekteish, Ziad Elnasser

**Affiliations:** 1Department of Family and Community Medicine, Faculty of Medicine, Jordan University, Amman 11942, Jordan; 2Department of Clinical Sciences, College of Medicine, Yarmouk University, Irbid 21163, Jordan; asmaa.mnayyis@yu.edu.jo; 3Department of Pathology and Microbiology, King Abdullah University Hospital, Jordan University of Science and Technology, Irbid 22110, Jordan; hmalzoubi6@just.edu.jo (H.A.); znasser@just.edu.jo (Z.E.); 4Medical Department, MENA Center for Research & Development and Internship, Amman 11931, Jordan; ruba.yousef1995@gmail.com (R.A.-A.); bat.fadi97@yahoo.com (F.B.); okkeh123@gmail.com (O.O.); abood.dalal.ad@gmail.com (A.D.); alhindia23@yahoo.com (A.A.); 5Ministry of Health, Princess Rahma Pediatrics Hospital, Irbid 21163, Jordan; jdaitawi_hussein@yahoo.com; 6Faculty of Medicine, Mutah University, Mutah 61110, Jordan; onafi2000@yahoo.com; 7Vaccines Department, Ministry of Health, Amman 11931, Jordan; d.abusal@yahoo.com; 8Royal Medical Services, Amman 1193, Jordan; altawalbeh@yahoo.com (A.A.); dralia.khlifat@gmail.com (A.K.); adoctor77@hotmail.com (M.A.); 9Department of Pediatrics, Faculty of Medicine, Jordan University, Amman 11942, Jordan; e.alzayadneh@ju.edu.jo; 10Pediatrics Department, Zarqa Governmental Hospital, Zarqa 13116, Jordan; ihsanalmaaitah66@gmail.com; 11Albashir Hospital, Amman 11931, Jordan; bob_borghul@hotmail.com; 12Department of Biomedical Engineering, School of Applied Medical Sciences, German-Jordanian University, Amman 11931, Jordan; adnan.lahham@gju.edu.jo; 13Department of Central Laboratories, Ministry of Health, Amman 11931, Jordan; 14Department of Pediatrics, Faculty of Medicine, King Abdullah University Hospital, Jordan University of Science and Technology, Irbid 22110, Jordan; faisal@just.edu.jo

**Keywords:** *Streptococcus pneumoniae*, serotype, Jordan, invasive pneumococcal disease, pediatrics

## Abstract

Introduction: *Streptococcus pneumoniae* infections are a major cause of mortality and morbidity worldwide. In Jordan, pneumococcal conjugate vaccines (PCVs) are not included in the national vaccination program. Due to the current availability of several PCVs, including PCV-10, PCV-13, and PCV-15, along with PCV-20, currently undergoing pediatric approvals globally, the decision to introduce PCVs and their selection should be based on valid local data on the common serotypes of *Streptococcus pneumoniae*. Methods: This cross-sectional study aimed to identify the frequency of serotypes of *Streptococcus pneumoniae* in children aged below 5 years hospitalized with invasive pneumococcal diseases (IPDs), including pneumonia, septicemia, and meningitis, during the study’s duration in representative areas of Jordan. Serotyping for culture-positive cases was based on the capsular reaction test, known as the Quellung reaction. qPCR was conducted on the blood samples of patients with lobar pneumonia identified via X-ray or on cerebrospinal fluid for those with a positive latex agglutination test for *Streptococcus pneumoniae*. Results: This study was based on the analysis of the serotypes of 1015 *Streptococcus pneumoniae* cases among children younger than the age of 5: 1006 cases with pneumonia, 6 cases with meningitis, and 3 cases with septicemia. Only 23 culture-positive cases were identified in comparison to 992 lobar pneumonia cases, which were PCR-positive but culture-negative, with a PCR positivity rate of 92%. Serotypes 6B, 6A, 14, and 19F were the most common serotypes identified in this study, with prevalence rates of 16.45%, 13.60%, 12.12%, and 8.18%, respectively. PCV-10, PCV-13, PCV-15, and PCV-20 coverage rates were 45.32%, 61.87%, 64.14%, and 68.47%, respectively. Discussion: To the best of our knowledge, this is the largest prospective study from the Middle East and one of the largest studies worldwide showing the serotypes of *Streptococcus pneumoniae*. It reveals the urgency for the introduction of a PCV vaccination in Jordan, utilizing recently developed vaccines with a broader serotype coverage.

## 1. Introduction

Pneumococcal disease describes a group of infections such as meningitis, pneumonia, septicemia, sinus infections, and ear infections caused by the *Streptococcus pneumoniae* bacterium [1]. *Streptococcus pneumoniae* infections constitute a major cause of mortality and morbidity worldwide. It is estimated that this bacterium contributes to more than one-third of the 2 million global annual deaths of children following acute respiratory infections (ARIs) [2]. The mortality rate from this bacterium varies considerably in reported studies, ranging from 7% to 36% [3,4,5]. Moreover, *Streptococcus pneumoniae* is the most common cause of community-acquired pneumonia requiring hospitalization, accounting for up to 50% of these cases [6]. Additionally, data from the US CDC revealed that this bacterium is the most common cause of pediatric infections for which antibiotics are routinely prescribed [7].

Three pneumococcal conjugate vaccines (PCVs) are currently available and widely used worldwide for the routine immunization of infants [8,9,10,11,12,13,14,15,16,17,18,19]. A summary of available PCVs is provided in Table 1. The first PCV used in a national pediatric immunization program (NIP) was PCV-7, which offers protection against seven serotypes (4, 6B, 9V, 14, 18C, 19F, and 23F) [8]. The 10-valent pneumococcal conjugate vaccine (PCV-10) offers seroprotection against three additional serotypes (1, 5, and 7F), not covered by PCV-7 [9], while the 13-valent pneumococcal conjugate vaccine (PCV-13) provides coverage for three additional serotypes (3, 6A, and 19A) not covered by PCV-10 [10].

Since the introduction of PCV-10 and PCV-13, there has been a worldwide change in the dominant *Streptococcus pneumoniae* serotypes, with a greater burden of non-PCV-13 serotypes [11,12,13]. Therefore, more vaccines have been developed or are under development to address the global changes in the epidemiology of this bacterium.

The recently developed PCV-15 provides coverage for two additional serotypes (22F and 33F) compared to PCV-13 [14]. This new vaccine has the potential to reduce morbidity and mortality from *Streptococcus pneumoniae* by providing a broader coverage for leading serotypes associated with pneumococcal disease worldwide [15]. The serotypes 22F and 33F unique to PCV-15 are among the leading serotypes causing IPD in children and adults following the widespread use of PCV-13 in children in many countries, likely due to their invasiveness capacity [16]. PCV-20 covers additional serotypes not included in PCV-15, including serotypes 8, 10A, 11A, 12F, and 15 [17]. The PCV-20 vaccine has been approved for adults [18]. Clinical trials are underway for the pediatric age group, with emerging promising safety and immunogenicity data [17].

In Jordan, vaccination against pneumococcal infections is not included in the national pediatric immunization program. This has been attributed to the limited resources and the lack of scientific evidence from Jordan to support vaccine introduction [19]. Additionally, there are no published data on the serotypes leading to IPD at representative sites locally. Due to the current availability of several PCVs [15], the decision to introduce the pneumococcal vaccine in the NIP and the method followed for vaccine selection should be based on objective valid local data on the common serotypes causing pneumococcal disease in Jordan.

## 2. Methods

This cross-sectional study aimed to identify the common serotypes of *Streptococcus pneumoniae* in children aged below 5 years hospitalized with invasive pneumococcal diseases (IPDs), including pneumonia, septicemia, and meningitis, during the study’s duration in representative areas of Jordan.

This study was based on data collected between 1 October 2021 and 31 December 2022 at Al-Bashir Hospital, Amman (1101 beds; 190 beds for pediatrics); King Abdullah I University Hospital, Irbid (750 beds; 86 beds for pediatrics); Queen Rania Al-Abdulla Hospital for Children, Royal Medical Services (212 beds); Princess Rahma Hospital for pediatrics (200 beds); Irbid, Zarqa Governmental Hospital, Al-Zarqa (568 beds; 52 beds for pediatrics); Jordan University Hospital, Amman (500 beds; 50 beds for pediatrics); Karak Teaching Hospital, Al-karak (174 beds; 53 beds for pediatrics); and Prince Rashid Ben Al-Hasan Military Hospital, Irbid (630 beds; 83 beds for pediatrics). The sites represent pediatric admissions from the central, northern, southern, and eastern parts of Jordan.

### 2.1. Case Definition [20,21]

A case of childhood IPD was defined as “a child with *Streptococcus pneumoniae* isolated from a normally sterile site, such as blood, cerebrospinal fluid (CSF), or pleural fluid” [5].

Lobar pneumonia was described as an acute exudative inflammation of an entire pulmonary lobe caused mainly by *Streptococcus pneumoniae* in more than 95% of cases [21]. Blood samples were collected from cases of lobar pneumonia based on clinical presentation and chest X-ray findings for PCR testing for *Streptococcus pneumoniae*, as described below.

### 2.2. Inclusion Criteria

All children younger than the age of 5 living in the study locations for more than 6 months diagnosed with an invasive pneumococcal infection (invasive pneumonia, septicemia, meningitis, etc.) and identified through culture results during the study duration at study sites were included.

*Streptococcus pneumoniae* contributes to more than 95% of lobar pneumonia cases [21]. Therefore, in addition to culture-positive cases at the study sites, blood samples were collected for bacteremia screening amongst children, and chest X-rays, with radiological findings suggestive of lobar pneumonia, were performed at the time of presentation. The molecular technique protocol for the identification of *Streptococcus pneumoniae* and then serotype specification is described below.

## 3. Exclusion Criteria

Children receiving routine pneumococcal vaccination as part of the private sector vaccination program or as high-risk groups. Only one case was excluded from the study due to receiving PCV-13 in a private clinic.Not permanently resident in the study area.

### Sample Size Calculations [22,23,24]

In Jordan, a PCV has not been introduced; therefore, the sample size should be based on data prior to PCV introduction. A conservative estimate of the global incidence of IPD before PCV vaccine introduction was 100 per 100,000 annually [25,26]. The required sample size to estimate the incidence of IPD and study *Streptococcus pneumoniae* serotypes locally was 1001 cases. This was estimated using the 95% significance level and error margins of 0.03 to be able to identify serotypes with the prevalence of 1 per 1000 (0.1%).

Statistical analysis plan: The collected data were analyzed using R Statistical Computing Software version 3.4.3 (R Foundation for Statistical Computing, Vienna, Austria). Descriptive statistics, including means, standard deviations, and confidence intervals, are reported for numerical patient characteristics, while frequencies and percentages are reported for categorical characteristics. The prevalence of different serotypes of pneumococcal infections based on the site of infection, age, gender, and geographical area is also reported. The chi-squared testing procedure was used to compare serotypes by site of infection. Predictors of the presence of certain serotypes of pneumococcal infections were identified through multinomial logistic regression models with stepwise selection. A significance level of 0.05 was used throughout the analysis [27].

## 4. Microbiology

As described below, two approaches were followed to identify *Streptococcus pneumoniae* for serotyping:PCR-positive cases: For cases with radiological findings suggestive of lobar pneumonia, as described in the radiology section below, blood samples were collected on admission. PCR was used to identify *Streptococcus pneumoniae* and for serotyping.Culture-positive cases: For invasive pneumococcal disease cases (pneumonia, meningitis, septicemia, etc.) identified during the study period at study sites, a Quellung test was performed for serotyping.

## 5. Sample Collection and PCR Analysis

Whole-blood samples were collected through venipuncture for PCR from patients with lobar pneumonia based on an X-ray or those with a CSF analysis suggestive of bacterial infection. Whole-blood samples were collected into an EDTA (ethylenediaminetetraacetic acid) tube from eligible candidates and kept refrigerated at 2–8 °C for a maximum of 3 days prior to nucleic acid extraction [28]. DNA extraction from whole blood or *Streptococcus pneumoniae* ATCC 49619 (positive control) bacterial broth was performed using a QIAamp DNA mini kit according to the manufacturer’s instructions (Qiagen, Hilden, Germany). The extracted DNA was kept frozen until it underwent PCR to detect the autolysin (lytA) gene. The detection of *Streptococcus pneumoniae* nucleic acid in whole blood using a qPCR assay was carried out based on the method from the US Centers for Disease Control and Prevention [29]. The reaction was performed at Mega Labs Amman, Jordan, using a BioQuant-96 (Biosan, Riga, Latvia) instrument with the following recommended cycling conditions: 95 °C polymerase activation for 10 min, 95 °C for 15 s, and 60 °C for 1 min for 40 cycles. A cycle threshold (CT) of <40 was considered “positive”.

In each run, *Streptococcus pneumoniae* ATCC 49619 was used as a positive control, and *Streptococcus pyogenes* ATCC 49399 was used as a negative control. A non-template negative control was also included to check for contamination. Positive samples were frozen until serotyping was performed using 20 pairs of primers that were previously validated (Table A1; real-time PCR assays with optimum conditions were utilized as described in references for each serotype primer). Although the primers were described and validated previously, an extra step was taken by the study team to confirm the results for the most common serotypes. Clinical isolates were used as controls for serotypes 6A, 6B, 14, and 19F from previous culture-positive cases identified through the Quellung reaction. Serotypes were then documented for each patient. Samples where no serotype was detected were marked as “other serotypes”. Serotyping was carried out using the Qiagen rotor gene instrument (Qiagen, Hilden, Germany). Primers were used to screen the samples for common serotypes included in the recently developed vaccines (PCV-15 and PCV-20) and for commonly reported serotypes not covered by these vaccines but reported post PCV-13 administration amongst hospitalized children, such as the following serotypes: 11 non-A subserotypes, 15 non-B subserotypes, 16, 18 non-C subserotypes, 22 non-F subserotypes, 28, 33 other subserotypes, 3F, 5 A, 6C/6D, and 7 non-F subserotypes [28,29].

## 6. Culture and Identification

Samples were processed at the local site, and then the *Streptococcal pneumonia* isolates were sent to the hospital central laboratory. Blood culture bottles were incubated in a BACT/ALERT automated microbial detection system (bioMérieux, Inc, Durham, USA). Positive samples were then cultured on 5% sheep blood agar (made selective by the addition of 5 µg/mL gentamicin) [30].

Confirmed pneumococcal isolates were stored in duplicate in 1.5 mL cryotubes containing STGG (skim milk–tryptone–glucose–glycerin) broth at −80 °C until being transported to the central laboratory for serotyping plus further characterization [31].

## 7. Serotyping for Culture-Positive Cases

The capsular reaction test, known as the Quellung reaction, is the gold standard method for pneumococcal typing, and it was first described by Neufeld in 1902 [32].

In this study, for culture-positive cases, the following steps were followed for serotyping using the Quellung reaction: After identifying the isolate as a pneumococcus, it was then tested with antiserum pools until a positive reaction was observed. Each pool of antiserum contained a different mixture of antisera raised against 91 pneumococcal serotypes. Once the pool was established, the individual type and group antisera that were included in the reactive pool were tested individually in sequence. A group antiserum reacts with all the serotypes in a particular group (e.g., group 22 antiserum reacts with serotypes 22F and 22A), whereas a type antiserum generally reacts with a single serotype (e.g., type 5 antiserum reacts with serotype 5) [33]. The serotyping of culture-positive cases was performed using a Pneumotest-Latex agglutination kit, and the results were confirmed by means of the Quellung reaction using serotype-specific antisera. All specimens were screened for multiple serotypes [32,33,34].

## 8. Radiological Findings

The diagnosis of community-acquired pneumonia in hospitalized children usually depends on the finding of consolidation (alveolar infiltrates) or consolidation complicated by effusion on a chest X-ray [35]. As described above, lobar pneumonia was described as an acute exudative inflammation of an entire pulmonary lobe and is caused mainly by *Streptococcus pneumoniae* in more than 95% of cases [36,37].

All X-rays for hospitalized children with pneumonia were screened initially by a consultant pediatrician-in-charge and then by a consultant radiologist from the study team (A.M.) to confirm the presence of lobar pneumonia. In cases of disagreement, the case was discussed, and a final judgment was made based on background and clinical data, including age, gender, medical and drug history, ICU admissions, and complications.

Leukocytosis was defined as a WBC count > 11.0 × 10^9^/L [38,39]. Complications during admission included pleural effusion, serious hypotension that resulted in severe hemodynamic changes, lung abscess, lung cavitation, sepsis with attendant shock, and acute respiratory failure [38,39].

Background and clinical data were obtained by the research coordinator at study sites under the supervision of the principal investigator at each site. The research coordinators were trained on data extraction. The PI at each site allocated two senior pediatric residents to assist the coordinators at each site in the clinical assessment and the extraction of clinical data from the electronic records.

## 9. Results

This study was based on the analysis of serotypes of 1015 *Streptococcus pneumoniae* cases. Lobar pneumonia was the final diagnosis for the majority of cases (1008, 99.3%). Based on the routine culture at the study site and during the study period, only 23 culture-positive cases were identified in comparison to 992 PCR-positive but culture-negative cases. All of these 23 culture-positive cases had similar results on the PCR for the blood or CSF fluid analysis. The results are consistent with the PCR findings: 6 cases were diagnosed with meningitis, 3 cases were diagnosed with sepsis, and the remaining 14 cases were diagnosed with pneumonia complicated by septicemia. Based on the admission reports at the study sites, lobar pneumonia contributed to 50% of all of the reported pneumonia cases. In total, 1078 blood samples were collected from cases with radiological findings suggestive of lobar pneumonia. The PCR positivity rate was 92%.

The baseline characteristics of the study sample are shown in Table 2. The mean age of the included cases was 15 ± 16.1 months. Around two-thirds of the cases were aged 2 years or younger (74.2%, N = 754). Overall, 42.5% of the sample were females, and 57.5% were males. Two-thirds of the participants (92.5%) were born at full term, compared to 7.5% with a reported history of preterm delivery. Overall, 32.8% of the sample was born through caesarian section, compared to 67.2% born through normal vaginal delivery. Smoking amongst family members was highly prevalent, with a rate of 47.3%. Most of the smokers reported smoking inside the home, with a rate of 91.3% of smokers, compared to 8.7% who smoked only outdoors.

The proportion of participants with congenital conditions was 7.7%. Congenital heart disease was the most commonly reported condition, with 43 cases out of the 78 patients having congenital conditions (5.5%). The prevalence of chronic illnesses, including congenital or acquired illnesses, was 12.6%. Bronchial asthma had the highest reported prevalence of 2.5%. Finally, 5.8% of the sample was on regular medications.

The clinical characteristics of the study participants (Table 2) show that 21.0% of the participants received antibiotics prior to admission. Overall, 241 cases (23.7%) in the sample required ICU admission; 95 of them required intubation. Leukocytosis was present in 74.9% of the samples [40]. Complications during admission were reported for 117 cases (11.5%) [38,39].

Figure 1 shows the distribution of *Streptococcus pneumoniae* serotypes based on the PCR analysis, including the 23 culture-positive cases.

Serotypes 6B, 6A, 14, and 19F were the most common serotypes identified in the study, with prevalence rates of 16.45%, 13.60%, 12.12%, and 8.18%, respectively. The same trend was also noted in children aged 2 years or younger, with the highest frequencies for serotypes 6B (15.65%) and 6A (14.06), followed by serotypes 14 (12.6%) and 19F (11.62%). The serotypes included in PCV-20 but not in PCV-13 contributed to 6.9% of the cases, with the highest rates for serotypes 12F, 11A, and 22F. These serotypes had frequencies of 1.99%, 1.86%, and 1.59%, respectively. Serotypes not covered by current vaccines, including PCV-20, contributed to 31.53% of the samples, among which none were dominant. The prevalence of these serotypes ranged from 0.1 to 1.08%.

As shown in Table A1, the serotypes included in PCV-10 contributed to 45.32% of reported serotypes, while the serotypes included in PCV-13 contributed to 61.87% of the cases. This is largely attributed to serotype 6A, with a frequency of 13.60%. Serotypes 3 and 19A that are present in PCV-13 but not PCV-20 contributed to 2.95% of the cases. The most recently approved vaccines—PCV-15 and PCV-20—provide a broader coverage for the serotypes detected in our study, with coverage rates of 64.14% and 68.47%, respectively, as shown in Figure 2. Similar results were obtained for children aged 2 years or younger with coverage rates of 45.23%, 61.54%, 63.79%, and 68.44% for PCV-10, PCV-13, PCV-15, and PCV-20, respectively.

Appendix A shows details of the Quellung test results for the 23 culture cases. These results are consistent with the PCR findings. Overall, 6 cases were diagnosed with meningitis, 3 cases were diagnosed with sepsis, and the remaining 14 cases were diagnosed with pneumonia complicated by septicemia.

A regression analysis included baseline characteristics as predictors of the coverage of the serotypes in PCV-10, PCV-13, PCV-15, or PCV-20. This included age, gender, region, mode of delivery, term of delivery being full term or preterm, the presence of congenital conditions, the presence of chronic illnesses, smoking at home, and being on regular medications. None of the studied predictors were statistically significant.

## 10. Discussion

To the best of our knowledge, this is the largest study from the Middle East and one of the largest prospective studies worldwide showing the serotypes of *Streptococcus pneumoniae* using molecular techniques through quantitative polymerase chain reactions (qPCRs) and the classical culture-based Quellung reaction. This study revealed the urgency for the introduction of PCV vaccinations in Jordan, utilizing vaccines with a broader serotype coverage, such as PCV-15 and PCV-20. PCV-13 provides a good coverage for the currently prevalent serotypes in Jordan (61.87%), while PCV-10 has limited use locally based on this study outcome, with a coverage rate of only 45.32% of identified serotypes.

In countries such as Jordan with no PCVs, *Streptococcus pneumoniae* continues to cause more hospitalizations leading to more morbidities, thus constituting an economic burden. The majority of cases were identified through qPCR for blood samples of patients with lobar pneumonia. Moreover, our study revealed consistent findings with global data, where *Streptococcus pneumoniae* contributed to 50% of community-acquired pneumonia amongst hospitalized children younger than the age of 5 [6]. Our data also revealed that relying on routine culture techniques will largely underestimate the true burden of *Streptococcus pneumoniae* infections and other bacterial infections, highlighting the importance of molecular techniques in the assessment of the burden of different pathogens in developing countries [41].

The burden of *Streptococcus pneumoniae* worldwide remains considerable. Although many countries worldwide introduced PCVs, the estimated global burden of pneumococcal lower respiratory tract infections was 44.7 million cases, along with 341,029 associated deaths occurring in children younger than five years of age in 2016, mainly attributed to non-PCV-13 serotypes [42]. To expand serotype coverage, PCV-15 with serotypes 22F and 3F, along with the serotypes within PCV-13, was developed and is currently undergoing approval in different countries worldwide [18]. Moreover, a 20-valent PCV (PCV20) containing PCV-13 components and seven additional serotypes (8, 10A, 11A, 12F, 15B, 22F, and 33F) has been approved for adult populations [14].

The newly covered serotypes were targeted based on several reasons, including generated epidemiological data on the prevalence of serotypes in different geographical areas post PCV-13 introduction [9,10,11,12,13,16,17,18], in addition to clinical data, such as disease severity, isolation amongst cases with IPD, and antibiotic resistance [43,44].

Serotypes 6B, 6A, 14, and 19F were the most common serotypes identified in this study. These findings are consistent with those of a recent study from Jordan that examined the prevalence of *Streptococcus pneumoniae* serotypes (nasopharyngeal colonization) in children in North Jordan [45]. The most prevalent *S. pneumoniae* serotypes were 6A/B (13.2%), 23F (7%), ST14 (6%), 9V (4.4%), 11A (3.2%), 19F (3%), ST3 (1.8%), ST4 (1.8%), 12F (1.6%), 35B (1.4%), 19A (0.8%), and 7F (0.6%). Our data are also consistent with regional and international data, particularly those reported prior to PCV vaccine introduction [45]. In high-income countries, serogroups 6, 9, 19, and 23 were among the most frequently identified in IPD [46]. The implementation of PCVs has led to a striking decrease in IPD incidence, with a decrease in the prevalence of vaccine serotypes (VTs) [9,10,11]. A recent systematic review studied the invasive disease potential of different serotypes: among non-PCV-13 serotypes, only serotype 12F seemed to have a higher disease potential than 19A, which is covered by PCV-13. Serotypes 8, 24F, and 33F were at the upper end of the invasiveness spectrum [9].

In this study, the serotypes included in PCV-20 but not in PCV-13 contributed to 6.9% of the cases, with the highest rates for serotypes 12F, 11A, and 22F. These data are lower than those reported from other countries because PCVs have not yet been introduced to the national immunization program in Jordan. Based on regional and international data, it is expected that the serotypes covered by PCV-20 but not by PCV-13 will dominate in Jordan if PCV-13 but not PCV-20 is utilized in the national program. This expectation is supported by a recent meta-analysis on serotypes leading to pediatric IPD following PCV introduction, which revealed that, post PCV-13 vaccine introduction, the most prevalent serotypes causing pediatric IPD were 22F (estimated 5.3% of cases), 12F (4.3%), 33F (4.5%), 15B (3.7%), 10A (3.4%), 8 (2.2%), and 11A (2.0%) [9]. Moreover, surveillance data obtained by the Centers for Disease Control and Prevention, USA, revealed that, in 2018, the seven new serotypes covered by PCV-20 alone accounted for an estimated 37% of IPD in children <5 years of age in the USA [44].

Our results and estimation of the local changes post PCV introduction in Jordan are also supported by regional data from the KSA pre- and post-vaccination introduction. Isolates from 250 blood and 61 cerebrospinal fluid samples were collected from different regions across KSA. The most frequently isolated IPD serotypes were 23F, 19F, 6B, 5, and 1 [10]. Over the course of the study and prior to PCV-13′s introduction, there was a significant rise in serotype 19A (covered by PCV-13 but not PCV7). There was a notable decrease in serotype 18C over this period, one of the PCV7 serotypes. The PCV-10 and PCV-13 coverage rates in this study period prior to PCV introduction were 80% and 90%, respectively.

Our study revealed that PCV-10 contributed to 45.32% of reported serotypes, while the serotypes included in PCV-13 contributed to 61.87% of the cases. This is largely attributed to serotype 6A, with a frequency of 13.60%, and to a lesser extent to 3 and 19A, which contributed to 2.95% of cases. The serotypes contained in existing vaccines continue to cause a substantial burden of IPD, including serotype 19A (particularly in countries using PCV-10 but not PCV-13 [12,17]. These data illustrate the limitations of relying on cross-protection for some serotypes (e.g., serotypes 19F for 19A) [13].

The newly developed PCV-15 and PCV-20 provide more coverage for the identified serotypes in Jordan, with coverage rates of 64.14% and 68.47%, respectively. The data reported by the World Bank from high-income countries support these findings. Overall, the average coverage of IPD was 10.7% (min: 0%, max: 51.5%) for PCV-10, 37.4% (min: 11.7%, max: 74.7%) for PCV-13, 44.5% (min: 15.0%, max: 79.6%) for PCV-15, and 66.5% (min: 42.3%, max: 93.1%) for PCV-20. A key finding from this analysis was that a high percentage of IPD cases were caused by the aggregate serotypes not included in PCV-10 and PCV-13 but included in PCV-15 and PCV-20 [47]. Similar data were obtained from Western Europe [48].

The above findings are of high importance for decision makers in Jordan to consider when selecting which PCV is to be introduced locally. The coverage of more serotypes will lead to more protection from *Streptococcus pneumoniae*, particularly for non-PCV-13 serotypes that are expected to dominate post PCV-13 introduction.

The assessment of the vaccines’ coverage via the presence of congenital illnesses or chronic illnesses did not show that these groups did not have serotypes specific to certain vaccines, such as those in PCV-15 and PCV-20. These findings were confirmed by a multiple regression analysis, which did not identify predictors for the coverage of different vaccines. This reveals that PCV-15 or PCV-20, covering more serotypes, should be applied for all children younger than the age of 5 years and not be limited to high-risk groups or as an optional choice for NIPs [49].

In summary, the distribution of serotypes of *Streptococcus pneumoniae* varies between countries and post PCV introduction. Local data, therefore, are needed to make evidence-based decisions on vaccine selection and to stimulate future research in the development of new vaccines [50]. The current efforts to introduce the PCV vaccine in Jordan are supported by our large number of PCR results but not culture-identified cases, along with our local data on the identified serotypes at representative sites in Jordan. Such data are of high importance, particularly in the presence of several approved vaccines for NIP evaluation [51,52,53,54,55].

In addition to the above data, evidence-based decisions on PCV introduction should be followed locally. These decisions include a cost effectiveness analysis, a budget impact analysis, an assessment of the cold chain, training plans, human resource capacity, political commitment to the introduction and sustainability of the vaccine, and public acceptance of the vaccine.

Our study has some limitations, the first of which is that 27.49% of the serotypes were labeled as “other serotypes”. The data were not based on cultures and were subject to the availability of primers and local resources for our project. The samples were stored at −80 °C, and the project team will consider further analysis for the unidentified serotypes. The second limitation of our study is that we limited our sample to invasive pneumococcal infections at sterile sites. Cases of otitis media, particularly complicated ones, such as perforated ear drums, were excluded. At some sites, cases were treated empirically at the emergency department without hospitalization. Efforts were made to ensure that these sites adhered to the local protocols of the admission of cases with lobar pneumonia. However, our study has two main advantages: The first one is that it is based on national representative samples from different regions. The second and main advantage of this study is that we presented the largest regional data and one of the largest prospective studies worldwide. Because of the use of molecular techniques (qPRCW), we presented serotypes for 1015 cases, when compared to 23 cases identified though the culture.

In conclusion, we recommend that the PCV vaccine should be immediately introduced in Jordan to control the growing burden of *Streptococcus pneumoniae*. PCV-20 and PCV-15 are the recommended vaccines of choice, followed by PCV-13. Developing countries need to depend on molecular techniques in identifying the burden of different infections and to avoid underestimating this burden when relying on culture results. This is a major issue in developing countries, particularly in the presence of antibiotics’ misuse. Finally, countries need to depend on local data in their NIP evaluation and updating due to variations between countries and regions in the burden of different infections and in the prevalence of different serotypes of the causative organism.

## Figures and Tables

**Figure 1 vaccines-11-01396-f001:**
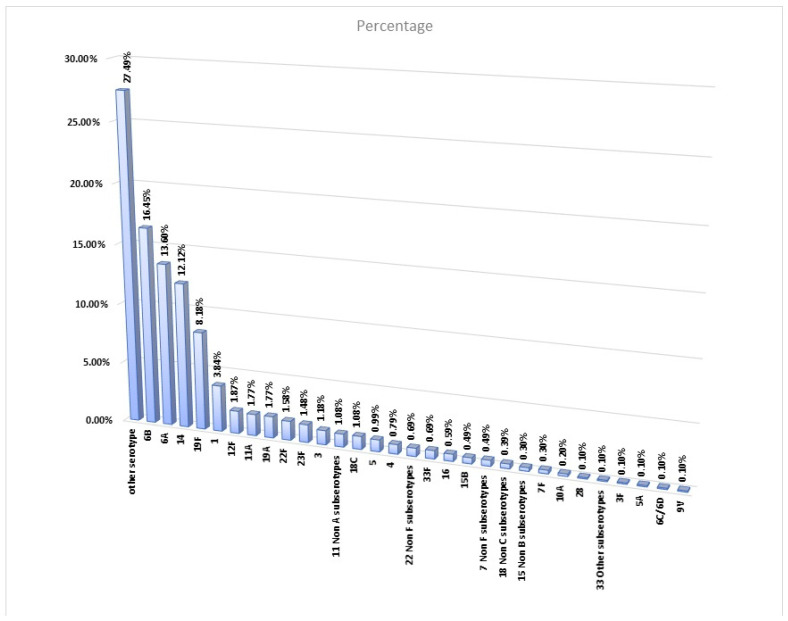
Frequency of detected serotypes for strep. pneumonia.

**Figure 2 vaccines-11-01396-f002:**
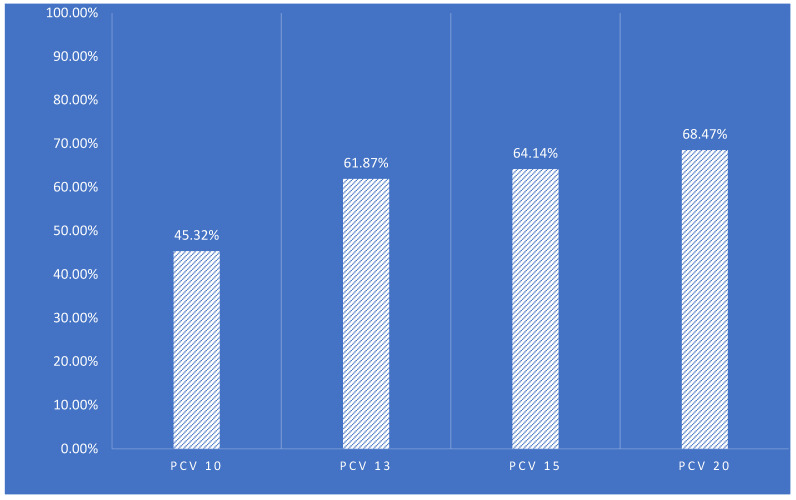
Coverage rates of local serotypes according to vaccine type (PCV-10, PCV-13, PCV-15).

**Table 1 vaccines-11-01396-t001:** Summary of current pneumococcal vaccines licensed for use.

Pneumococcal Vaccine	Covered Serotypes
PCV-7 [8]	4, 6B, 9V, 14, 18C, 19F, and 23F
PCV-10 [9]	PCV-7 plus 1, 5, and 7F
PCV-13 [10]	PCV-10 plus 3, 6A, and 19A
PCV-15 [14]	PCV-13 plus 22F and 33F
PCV-20 [17]	PCV-15 plus 8, 10A, 11A, 12F, and 15

**Table 2 vaccines-11-01396-t002:** Participants clinical and demographic characteristics (N = 1015).

Variable	Category	N (%)
Sex	Male	584 (57.5%)
	Female	431 (42.5%)
Birth history	Full term	937 (92.3%)
	Preterm	78 (7.7%)
Mode of delivery	Normal VD	683 (67.3%)
	Caesarean section	332 (32.2%)
Congenital conditions	Yes	78 (7.7%)
Congenital heart disease	Yes	43 (4.2%)
Chronic illness	Yes	128 (12.6%)
Asthma	Yes	25 (2.5%)
Family members’ smoking place	Inside home	438 (43.2%)
	Only outside home	42 (4.1%)
	Not smoker	535 (52.7%)
Patient received antibiotics within a week of admission	Yes	213 (21.0%)
Previous hospitalization	Yes	220 (21.7%)
Patient receives regular medications	Yes	59 (5.8%)
Fever	Yes	715 (70.4%)
Blood culture	Positive	23 (2.2%)
	Negative	987 (97.3%)
	Not performed	5 (0.5%)
Complications during admission	Yes	117 (11.5%)
Required ICU admission	Not admitted	774 (76.2%)
	Admitted and intubated	95 (9.4%)
	Admitted and not intubated	146 (14.4%)
WBC (leukocytosis)	Negative	255 (25.1%)
	Positive	760 (74.9%)

## Data Availability

All data are referred to in the main text. Any further data or clarifications can be provided through the corresponding author.

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
