# Peer review of "Epidemiology of *Streptococcus pneumoniae* Serotypes in Jordan Amongst Children Younger than the Age of 5: A National Cross-Sectional Study"

_vaccines, 2023, doi:10.3390/vaccines11091396_

Round 1
Reviewer 1 Report
Dear Editor,
The MS entitled “Epidemiology of Streptococcus pneumonia Serotypes in Jordan 2 Amongst Children Younger than the Age of 5: a National 3 Cross-Sectional Study” describes the urgency for the introduction of PCV vaccination in Jordan, 61 utilizing recently developed vaccines with broader serotype coverage. The study describes the introduction of pneumococcal vaccine in the national pediatric immunization program which is the need.
- The methodology is too long. It can be shortened.
- In result section, legends can be improved. There is no legend for figure 1. Figure 2 can be improved.
- Legends for Tables must be improved. There is no legend for Table 4.
- Discussion can be improved by citing the figures and tables in the discussion section.
Dear Editor,
The MS entitled “Epidemiology of Streptococcus pneumonia Serotypes in Jordan 2 Amongst Children Younger than the Age of 5: a National 3 Cross-Sectional Study” describes the urgency for the introduction of PCV vaccination in Jordan, 61 utilizing recently developed vaccines with broader serotype coverage. The study describes the introduction of pneumococcal vaccine in the national pediatric immunization program which is the need.
- The methodology is too long. It can be shortened.
- In result section, legends can be improved. There is no legend for figure 1. Figure 2 can be improved.
- Legends for Tables must be improved. There is no legend for Table 4.
- Discussion can be improved by citing the figures and tables in the discussion section.
Author Response
Thank you very much for your great efforts and valuable input. Kindly find below our feedback on below comments.
- The methodology is too long. It can be shortened.
Thank you! 12 lines have been removed from the methodology. We kept the remaining parts to ensure that the reader know that we followed valid techniques and to help others in the future to follow the same protocol for similar work.
- In result section, legends can be improved. There is no legend for figure 1. Figure 2 can be improved.
Thank you. This has been fixed. Both figures have been edited.
- Legends for Tables must be improved. There is no legend for Table 4.
Thank you, This has been fixed.
- Discussion can be improved by citing the figures and tables in the discussion section.
I agree that this could be helpful, however, we have followed the journal style for keeping figures and tables to the results section.
Reviewer 2 Report
The present work has scientific relevance for addressing a highly prevalent disease worldwide. The dissemination of Streptococcus pneumoniae serotypes, as well as the epidemiological characterization of children in Jordan, contributes to the adoption of public measures. However, for a better visualization of the results, some points need to be improved:
1. Table 1 can be taken from the introduction section.
2. The authors could only include the positive result (Yes) of the categorical variables (such as congenital heart disease), in the column (category).
3. Table 3 could be presented in the appendix section.
4. Table 4 could be included in text form only.
Author Response
Thank you very much for your great efforts and valuable input. Kindly find below our feedback on below comments.
- Table 1 can be taken from the introduction section.
We are happy to remove but thought it could help by giving readers a quick comparison between vaccines in the serotypes coverage. We have kept it and willing to remove, if highly needed.
- The authors could only include the positive result (Yes) of the categorical variables (such as congenital heart disease), in the column (category).
Thank you very much! We have done so!
- Table 3 could be presented in the appendix section.
We are happy to move as an appendix, if needed. We thought its presence in the text would be of more value.
- Table 4 could be included in text form only.
I see your feasible point. We have,therefore, transferred it into a supplementary table. We have kept it because many developing countries including Jordan are counting only on culture in assessment of burden of bacterial infections leading to underestimation of the true burden strep. Pneumonia and others infection.
Best regards
Reviewer 3 Report
Estimated Authors,
I've read with great interest the present paper on the epidemiology of S pneumoniae (pneumococcus) in the middle-eastern country of Jordan. Authors have provided an accurate reporting on a total of 1015 cases of pneumococcal infection, and corresponding serotypes.
Authors have also provided an appropriate appraisal of the circulating serotypes for envisaging the more appropriate vaccination strategy and avoiding the imbalance between vaccine and actual pathogens.
The present paper, in the end, confirms that the difference between PCV-13, PCV-15 and PCV-20 in terms of actual coverage of the general population could lead to substantial debate, with the eventual decision for the more appropriate vaccination strategy having more a health politics nature than an evidence based one.
I've only a couple of notes.
The first one, very simple: please fix the main title: pneumoniae, not pneumonia.
The second one: International readers should be briefed about the nature of the healthcare provided to Jordan children. As the study was based on the specimens of "children aged below 5 years hospitalized with invasive pneumococcal diseases (IPD) including pneumonia, septicemia and meningitis", any selection bias may lead to improper estimates. Could you rule any kind of self-selection of patients based on the severity and economic status of participants? in this regard, have you noticed any geographic gradient in the typization?
Thank you for this very interesting study.
The overall quality of the English text is appropriate. Only some minor typos scattered across the main text.
Author Response
Thank you very much for your great efforts and valuable input. Kindly find below our feedback on below comments.
The first one, very simple: please fix the main title: pneumoniae, not pneumonia.
Thank you. This was unintentional mistake.
The second one: International readers should be briefed about the nature of the healthcare provided to Jordan children. As the study was based on the specimens of "children aged below 5 years hospitalized with invasive pneumococcal diseases (IPD) including pneumonia, septicemia and meningitis", any selection bias may lead to improper estimates. Could you rule any kind of self-selection of patients based on the severity and economic status of participants? in this regard, have you noticed any geographic gradient in the typization?
Thank you very much for this valid and important point. We clearly stated in the inclusion criteria that all eligible children were screening. The study, as stated in the methodology, was conducted at a national level at all region. In the regression analysis, we included “region” in the predictors of predictors of coverage of serotypes in PCV-10, PCV-13, PCV-15 or PCV-20, with no statistically significant differences.
Best regards